# Biophysical characterization of the CXC chemokine receptor 2 ligands

Patrick Martin[1], Emily A. Kurth[1], David Budean[1], Nathalie Momplaisir[2], Elaine Qu[2], Jennifer M. Simien[1], Grace E. Orellana[1], Chad A. Brautigam[3], Alan V. Smrcka[2], Ellinor Haglund[1]*

1 Department of Chemistry, University of Hawaii at Manoa, Honolulu, Hawaii, United States of America,
2 Department of Pharmacology, University of Michigan Medical School, Ann Arbor, Michigan, United States of America, 3 Department of Biophysics and the Department of Microbiology, University of Texas Southwestern Medical Center, Dallas, Texas, United States of America

☉ These authors contributed equally to this work.
* ellinorh@hawaii.edu

**Data Availability Statement:** All relevant data are within the manuscript and its Supporting information files.

**Funding:** National Science Foundation (NSF) award number CHE2145906 the Hawaii Community

## Abstract

The chemokines of the immune system act as first responders by operating as chemoattractants, directing immune cells to specific locations of inflamed tissues. This promiscuous network is comprised of 50 ligands and 18 receptors where the ligands may interact with the receptors in various oligomeric states i.e., monomers, homodimers, and heterodimers. Chemokine receptors are G-protein coupled receptors (GPCRs) present in the membrane of immune cells. The migration of immune cells occurs in response to a concentration gradient of the ligands. Chemotaxis of neutrophils is directed by CXC-ligand (CXCL) activation of the membrane bound CXC chemokine receptor 2 (CXCR2). CXCR2 plays an important role in human health and is linked to disorders such as autoimmune disorders, inflammation, and cancer. Yet, despite their important role, little is known about the biophysical characteristics controlling ligand:ligand and ligand:receptor interaction essential for biological activity. In this work, we study the homodimers of three of the CXCR2 cognate ligands, CXCL1, CXCL5, and CXCL8. The ligands share high structural integrity but a low sequence identity. We show that the sequence diversity has evolved different binding affinities and stabilities for the CXC-ligands resulting in diverse agonist/antagonist behavior. Furthermore, CXC-ligands fold through a three-state mechanism, populating a folded monomeric state before associating into an active dimer.

## Introduction

Chemokines are a group of small signaling proteins that play a crucial role in immune system regulation and cell migration. They are part of the cytokine family, which are secreted proteins involved in cell-to-cell communication [1]. Acting as chemoattractants, they attract immune cells to specific locations in the body. By controlling the migration of immune cells in response to chemokine gradients, the body can direct the appropriate immune cells to specific locations within tissues, ensuring an efficient and targeted immune response [2–4]. This mechanism is

Foundation (HCF) award number HCF40846 (013357-00002) The funders had no role in study design, data collection and analysis, decision to publish, or preparation of the manuscript.

**Competing interests:** The authors have declared that no competing interests exist.

crucial for various physiological processes, including immune surveillance, inflammation, and wound healing. Chemokines are produced by various cell types, including immune cells, endothelial cells, and fibroblasts. When tissue damage or infection occurs, immune cells release chemokines to recruit other immune cells to the affected area, which is essential for the proper functioning of the immune system [2–4]. Chemokines induce signal transduction by binding to specific G-protein coupled receptors (GPCRs) on the surface of immune cells [5]. When a chemokine ligand binds to its cognate receptor, it triggers a phosphorylation cascade of intracellular signaling events that regulate cell migration, adhesion, and activation. The CXC Receptor 2 (CXCR2) may activate six different signaling pathways involving: (*i*) Phosphoinositide 3-kinase (PI3K), (*ii*) Akt (Protein Kinase B), (*iii*) Mitogen-Activated Protein Kinase (MAPK), (*iv*) Small GTPases, (*v*) Nuclear Factor-Kappa B (NF-κB), and (*vi*) cAMP/PKA Pathways [6].

## CXC chemokine ligands

Chemokines are classified into four different classes characterized by the arrangement of their cysteine residues, i.e., CXC, CC, C, and $CX_3C$. Chemokine ligands have a conserved region consisting of an N-terminal region followed by three antiparallel β-strands and a C-terminal α-helix [7] (Fig 1). The interaction between the ligands and the cognate receptors primarily involves a conserved sequence of three amino acids (ELR) in an unstructured region of the N-terminus of the ligands and an extracellular loop of the receptor [7]. The ligands share high structural integrity but low sequence identity. Interestingly, chemokine-ligands may activate the receptors as monomers, homodimers, and/or heterodimers [5]. Despite the important role of chemokines for the immune system, little is known about the biophysical properties of chemokine ligands.

The biophysical characteristics of proteins, including the process of folding into their three-dimensional structure and their conformational dynamics, are intimately linked to their structure-function relationships. These relationships are crucial in controlling cellular processes

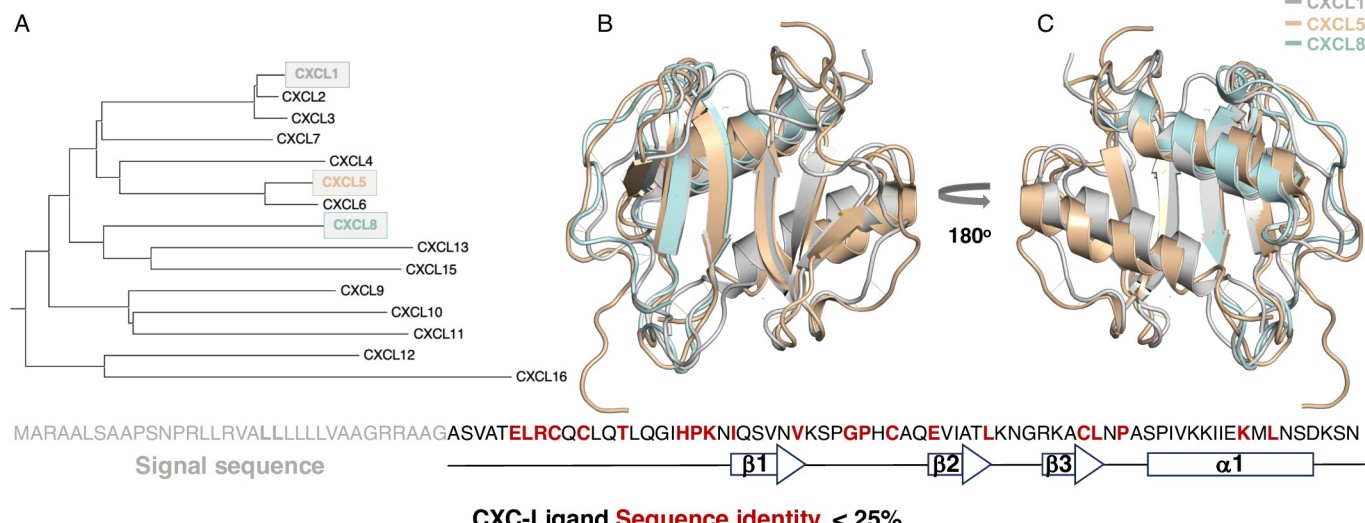

**Fig 1. The CXC-ligands (PDB ID 1MGS, 2MGS, and 1IL8).** A) A phylogenetic tree for CXC-chemokines generated from Uniprot [41]. B) and C) An overlay of the CXCL1, CXCL5, and CXCL8 structure and the sequence identity depicting the identical amino acids in red, including the ELR region that binds to CXCR2, for all three sequences. B) depicts the front view and C) the back view.

and signaling pathways important to human health [8–10]. It is well established that the three-dimensional structure of a protein does not determine its folding mechanism, as many proteins with conserved structures fold with different mechanisms [11–15]. In this work, we utilize the promiscuous chemokine ligands for CXCR2 [16]. The CXC-ligands (CXCL) for CXCR2 have a low sequence identity, < 25%, containing a conserved ELR region at the N-terminus essential for receptor interaction (Fig 1).

Despite the low sequence identity, the CXC-ligands share a structural homology. Using the pairwise Structure Alignment from the PDB (jFATCAT (rigid)), CXCL1-CXCL8 and CXCL1-CXCL5, respectively, have a Root Mean Square Deviation (RMSD) score of 2.31 and 2.98 and a Template Modeling score (TM-score) of 0.41 and 0.78 [17]. To investigate the biophysical characters of the CXC-ligands, we conducted thermodynamic and kinetic experiments.

To provide a foundation for understanding the promiscuity of the chemokine network, we investigate three of the CXCR2 ligands, namely CXCL1, CXCL5, and CXCL8. The ligands were strategically selected from sequence evolution representing different branches of the phylogenetic tree (Fig 1A). The CXCR2 receptor has seven cognate ligands that may interact and initiate a cell signaling response. CXCL1, CXCL5, and CXCL8 were selected to represent the most diverse ligands for CXCR2 that share a common ancestor. Our results show that the dimer of the CXC-ligands fold with a three-state mechanism, populating a fully folded monomeric state before associating into an active homodimer. CXCL8 is the most stable dimer with a maximal ERK phosphorylation at a lower concentration relative to CXCL1 and CXCL5. We hypothesize that sequence diversity evolved to induce specificity for protein:protein interaction leading to a promiscuous network to control the biological response.

## Materials and methods

### Cloning, expression, and purification

The CXCL5 (PDB ID: 2MGS) target gene cloned into a pET-32a Xa/LIC vector was provided by Krishna Rajarathnam, from the University of Texas Medical Branch at Galveston and the Department of Biochemistry and Molecular Biology [7]. The CXCL1 (PDB ID: 1MGS) and CXCL8 (PDB ID: 1IL8) target genes were designed similar to CXCL5 and cloned into a pET-32a Xa/LIC vector (Genscript). The CXCL1$^{I58W}$ and CXCL5$^{F62W}$ substitutions were constructed through site-directed mutagenesis using primers (Integrated DNA technologies, IDT) and the Quick-change site-directed mutagenesis kit (Stratagene). All proteins were recombinantly expressed as a fusion protein from the pET-32 Xa/LIC vector. The DNA vector was transformed into *E. coli* BL(21)DE3 cells (Agilent) and cultured in Luria Broth (LB) media. Transformed cells were grown to an $OD_{600} \sim 0.6$ and induced with 0.4 mM IPTG for 16 hours at 25 ˚C. The fusion proteins from the lysed cells were purified from inclusion bodies. The pellet was solubilized in 6 M GdmCl, 10 mM Tris, 1 mM EDTA, 0.1 mM DTT, 1 mM PMSF, pH 8 for 1 hour and refolded overnight in 10 mM Tris, 0.2 mM of oxidized glutathione, 2 mM of reduced glutathione, 1 mM EDTA, at pH 8 at 4 ˚C. The solubilized refolded fusion-protein was purified using SPFF ion-exchange chromatography, buffer exchanged to optimal cleavage conditions (10 mM Tris, 50 mM sodium chloride, 2 mM calcium chloride, pH 8), and the fusion tag was cleaved using Factor Xa protease (NE Biolabs & Sigma Aldrich). The cleaved CXC-ligands were purified using a high-prep SPFF ion-exchange column followed by S-100 size-exclusion chromatography (Cytiva). The purity of the protein was analyzed by SDS-PAGE and MALDI-TOF (S1 Fig in S1 File). Analytical ultracentrifugation (AUC) was conducted to select protein concentration at which the dimer is populated (S2 Fig in S1 File).

## Thermodynamic experiments

Equilibrium titration experiments were conducted utilizing a Chirascan v100 spectrometer (Applied Photophysics, U.K.). The secondary structure formations were probed by collecting circular dichroism (CD) spectra in far UV (240–190 nm) at a protein concentration from 2 to 100 μM using a 1 mm cuvette. Equilibrium titration measurements were collected by using CD to monitor the fraction of denatured protein between 225–219 nm in 0 to 8 M GdmCl in 10 mM potassium phosphate buffer, pH 7.4 at 25 ˚C. The change in Gibbs free energy (ΔG) is quantified using the equation:

$$\Delta G_{D-N} = -RTlnK = -2.3RTlogK \tag{1}$$

where the equilibrium constant ($K$) is the ratio between the denatured state [D] and the native state [N], R is the gas constant, and T is the temperature in Kelvin.

The fraction of the observed species ($F_{app}$) is represented by a two-state fit [18] shown by:

$$F_{app} = \frac{Y_N - Y}{Y_N - Y_D} \tag{2}$$

where the Y is the CD signal for the specified species, the two fractions [D] and [N]. The observed CD signal is plotted against denaturant concentration. The fitting equation for a two-state model:

$$Y = Y_N f_N + Y_D f_D \tag{3}$$

The thermodynamics data of CXC-ligands are also fitted to a three-state reaction according to $2D \rightleftharpoons 2N_{Monomer} \rightleftharpoons N_{Dimer}$ where the equilibrium constant K is:

$$K_{D-N_{Monomer}} = \frac{[N_{Monomer}]}{[D]} \; ; \; K_{N_{Monomer}-N_{dimer}} = \frac{[N_{Dimer}]}{[N_{Monomer}]^2} \tag{4}$$

where $K_{D-N_{Monomer}}$ represents the equilibrium constant between the denatured state [D] and native state [N_monomer]. $K_{N_{Monomer}-N_{Dimer}}$ represents the equilibrium constant between the native monomer [N_Monomer] and native dimer [N_Dimer]. The fraction of the observed species is represented by a three-state fit [18] shown by:

$$F_{app} = \frac{Y_{N_{Dimer}} - Y_{N_{Monomer}} - Y}{Y_{N_{Dimer}} - Y_D} \tag{5}$$

The fitting equation for a three-state model:

$$Y = Y_{N_{Dimer}} f_{N_{Dimer}} + Y_{N_{Monomer}} f_{N_{Monomer}} + Y_D f_D \tag{6}$$

## Kinetic experiments

Chevron plots for CXC-ligands were obtained from kinetics experiments utilizing tryptophan fluorescence conducted on a SX20 stopped-flow spectrometer (Applied Photophysics, Leatherhead, U.K.). The excitation wavelength was 280 nm using an LED source and emission was collected at 320 nm using a 295 nm cut-off filter. The experiment was conducted utilizing a 1/11 dilution with asymmetric mixing in the cuvette. The final protein concentration in the cuvette is between 2 to 30 μM utilizing the same conditions as for thermodynamics experiments. To solve the complex kinetics, 3.0, 2.7, and 3.5 M GdmCl were added to the unfolding experiments for CXCL1[I58W], CXCL5 [F62W], and CXCL8 respectively.

The kinetics data of both the fast phase and the slow phase were fitted with a two-state model according to:

$$\log k_{obs} = \log(k_f + k_u) = \log[10^{(\log k_f^{H_2O} + m_f(GdmCl))} + 10^{(\log k_u^{H_2O} + m_u(GdmCl))}] \tag{7}$$

where $k_f^{H_2O}$ and $k_u^{H_2O}$ are the refolding and unfolding rates in water. The $m_{D-N}$ is the solvent exposed surface area calculated from the slopes of the refolding limb ($m_f$) and the unfolding limb ($m_u$) of the chevron plots [19]. The linear relationship between the concentration of denaturant and the logarithmic function of the rate of folding is graphically represented as a chevron plot.

## Cell culture and transfection

HEK293 cells (CRL-1573, ATCC) were maintained in DMEM (10013 CV, Corning) supplemented with 10% FBS (10437028, Gibco) and 1% Pen/Strep (15140122, Gibco) at 37 ˚C with 5% $CO_2$. Transfection using lipofectamine 2000 (11668027, Thermo Scientific) was adapted from the manufacturer's protocol. In a 100 mm dish, $1.5 \times 10^6$ cells were transfected with 1 mg of 3xHA-CXCR2 (#CCR020TN00, cDNA) at a ratio of 1 mg DNA: 3 mL Lipofectamine 2000. Following transfection, media was supplemented with 500 mg/mL geneticin (G8168, Sigma) for the selection of HEK293-CXCR2 cells.

## ERK phosphorylation

HEK293-CXCR2 cells were plated in a 6 well-plate at a density of 40,000 cells per well in reduced serum media (0.5% FBS). Approximately 24 hours later, cells were serum-starved in DMEM supplemented with 1% Pen/Strep at 37 ˚C with 5% $CO_2$ for 2 hours. Following serum starvation, cells were stimulated with indicating concentrations of CXC-ligands for 5 mins and stopped with ice-cold PBS. Subsequently, 200 mL of 1 x Laemmli buffer was added per well. Lysates were collected, resolved on 4 to 20% Mini-protean TGXTM Gel (4561094, Bio-Rad) and transferred onto a nitrocellulose membrane (1215458, GVS). After transfer, the membrane was incubated with TBS buffer supplemented with 0.1% Tween 20 (TBS-T) and 3% BSA at room temperature for 1 hour on a shaker. The membrane was probed with primary antibodies (anti-phosphoERK1/2 (Cell Signaling 4370S)), anti- anti-ERK1/2 (Cell Signaling 4696S) diluted 1:1000 in TBS buffer supplemented with 0.1% Tween 20 and 3% BSA at 4 ˚C overnight. The next day, the membrane was washed with TBS-T four times and probed with the secondary antibodies (goat anti rabbit IgG, 680 RD (926–68071, LiCor), IRDye® 800RD Goat anti-Mouse IgG (926–32210, LiCor) at 1:10,000 dilution at room temperature for 1 hour. After another four washes with TBS-T, protein bands on the membrane were visualized using Li-Cor Odyssey CLx and analyzed using StudioLite software [20].

## Results

Understanding the three-dimensional structure and the thermodynamic and kinetic behavior of proteins is essential to understanding protein-protein interaction. The first structure of CXC-chemokines was solved about 30 years ago. However, the biophysical characters are not known for CXC-ligands. In this work, we investigate the molecular details controlling protein-protein interaction to understand the structural integrity controlling the oligomeric state of CXC-chemokines.

## Thermodynamics experiments

Chemokines are active molecules in various monomeric and dimeric states [5]. The dimerization depends on protein concentration, salt concentration, and pH [21–23]. We investigate the thermodynamic behavior of the homodimer formation of CXCL1, CXCL5, and CXCL8. The equilibrium curves display a slightly bimodal behavior, especially for CXCL8, with a shift both at low and high denaturant concentration, making the interpretation of the data more complex (Fig 2). The shape of the equilibrium curve of a three-state mechanism depends on the stability of the intermediate state during the folding process. Dimeric proteins display a concentration dependence, where the monomeric state becomes more populated at low concentrations while at high concentrations the dimeric state becomes more populated. Therefore, we tested three different protein concentrations (S3 Fig in S1 File). The data shows similar curves at high concentrations, from 30 to 100 μM, while there is a shift at 2 μM at low denaturant concentrations (S3 Fig in S1 File). This indicates that we have a monomeric intermediate state, which corroborates our hypothesis that the CXC-ligands fold through a three-state mechanism from $2D \rightleftharpoons 2N_{Monomer} \rightleftharpoons N_{Dimer}$ populating a folded monomeric ($N_{Monomer}$) intermediate state before populating the dimeric state ($N_{Dimer}$). However, the equilibrium titration curve of CXCL1 and CXCL5 does not display a significant bimodal behaviour compared to CXCL8 (Fig 2 and S3 Fig in S1 File). When the curves appear to be two-state, this phenomenon occurs at specific concentrations where only the native dimer and the unfolded monomer will be significantly populated. In this case, fitting the data to a two-state model will give the same $m$-value and Gibbs free energy as a three-state model for the folding transition. Thus, we fit the data both to a three-state fit (Table 1) and a two-state fit (S1 Table in S1 File). For CXCL1, there is one transition state at high denaturant concentrations with a ΔG of 5.40 ± 0.30 kcal/mol and a midpoint (MP) at 3.74, when fitted to a three-state equation. When fitted to a two-state equation, from $D \rightleftharpoons N$, ΔG is equal to 3.68 ± 0.03 kcal/mol with a MP at 3.31. For CXCL5, the stability for the first and second transition state is equal to a ΔG of 1.11 ± 0.29 and 6.94 ± 0.20 kcal/mol with a MP at 2.13 and 3.21 at low and high denaturant concentrations. When fitted to a two-state transition, ΔG is equal to 5.57 ± 0.04 kcal/mol with a MP at 3.32. For CXCL8, the stability for the first and second transition state is equal to a ΔG of 1.82 ± 5.27 and 7.60 ± 0.06 kcal/mol with a MP at 2.28 and 4.94 at low and high denaturant concentrations. When fitted to a two-state transition, ΔG is equal to 7.07 ± 0.10 kcal/mol with a MP at 5.00. The m-values for the two-state fit captures the unfolding transition at high denaturant and is in agreement with the m-value for the folding transition at high denaturant concentrations, $m_{D-N}^2$ from a three-state fit (Table 1 and S1 Table in S1 File).

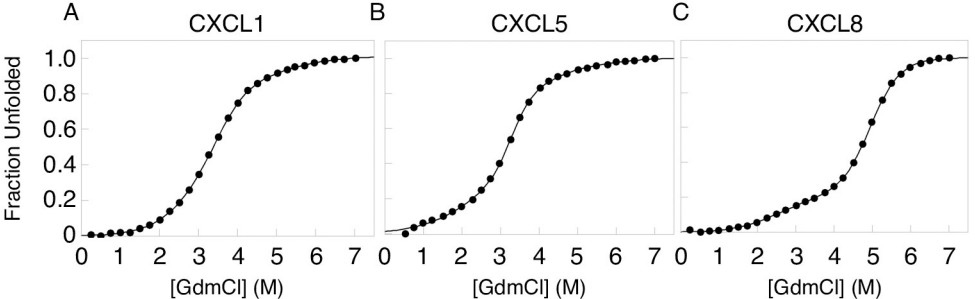

**Fig 2. Thermodynamics data of CXC-ligands.** The data for A) CXCL1, B) CXCL5, and C) CXCL8 are fitted to a three-state equation.

**Table 1. Equilibrium titrations in 10 mM phosphate buffer at pH 7.4 at 25 ˚C.**

|  | $MP^2$ M | $m_{D-N}^2$ $M^{-1}$ | $\Delta G^2$ kcal/mol | $MP^1$ M | $m_{D-N}^1$ $M^{-1}$ | $\Delta G^1$ kcal/mol |
|---|---|---|---|---|---|---|
| [3]CXCL1 | 3.74 | 1.06 | 5.40 ± 0.30 | n.d. | n.d. | n.d. |
| CXCL1[I58W] | 4.33 | 1.02 | 6.01 ± 0.15 | 3.08 | -0.38 | 1.58 ± 0.21 |
| CXCL5 | 3.21 | 1.59 | 6.94 ± 0.20 | 2.13 | -0.38 | 1.11 ± 0.29 |
| CXCL5[F62W] | 4.43 | 0.61 | 3.67 ± 0.26 | 1.42 | -0.60 | 1.58 ± 3.55 |
| CXCL8 | 4.94 | 1.13 | 7.60 ± 0.06 | 2.28 | -0.59 | 1.82 ± 5.27 |

*Data fitted to a three-state equation.

[1]Data fitted for transition one at low GdmCl.

[2]Data fitted for transition two at high GdmCl.

[3]CXCL1 shows a two-state behavior due to the concentration used, thus values not determined (n.d.).

## Kinetics experiments

The kinetics experiments display an atypical behavior of a fast and a slow phase at different time scales (Fig 3). This agrees with previously published data for CC-ligands [24]. The fast phase of the CXC-ligands shows a refolding rate of log $k_f^{H_2O} = 2.27$, 1.65, and 3.56 for CXCL1[I58W], CXCL5[F62W], and CXCL8 (Table 2). The surface exposed area is within the experimental error of the folded monomeric structures, with an $m_{D-N}$ value from 0.73 to 0.81 for CXCL5[F62W] and CXCL8. The compactness of the Transition State Ensemble (TSE) is calculated between 0–1 using the Tanford β-value (β‡) in the equation $\beta^\ddagger = \frac{m_f}{m_{D-N}}$, where a TSE resembling the denatured state has a low value and a TSE resembling the native state has a high value [19]. The β‡ for the CXC-ligands are 0.30, 0.24, and 0.44, for CXCL1[I58W], CXCL5[F62W], and CXCL8 respectively, which indicates a diffuse TSE resembling the denatured state with few native contacts formed. The MP shows a small shift from 4.58, 4.99, and 5.37 for CXCL1[I58W], CXCL5[F62W], and CXCL8, as well as the global stability $\Delta G = 4.96 \pm 0.69$, 4.47 ± 2.58, and 5.93 ± 1.82 kcal/mol for CXCL1[I58W], CXCL5[F62W], and CXCL8.

The refolding rate of the slow phase, log $k_f^{H_2O}$, is -0.12 and -0.04 for CXCL5[F62W] and CXCL8 (Table 3). There is a significant shift of the surface exposed area with an $m_{D-N}$ of 2.90 and 5.75 for CXCL5[F62W] and CXCL8. We attribute the large change to the difference in dimerization between the ligands, where CXCL1[I58W] dominantly is present as a monomer, CXCL8 as a dimer, and CXCL5[F62W] is somewhere in between at the low protein concentrations used for

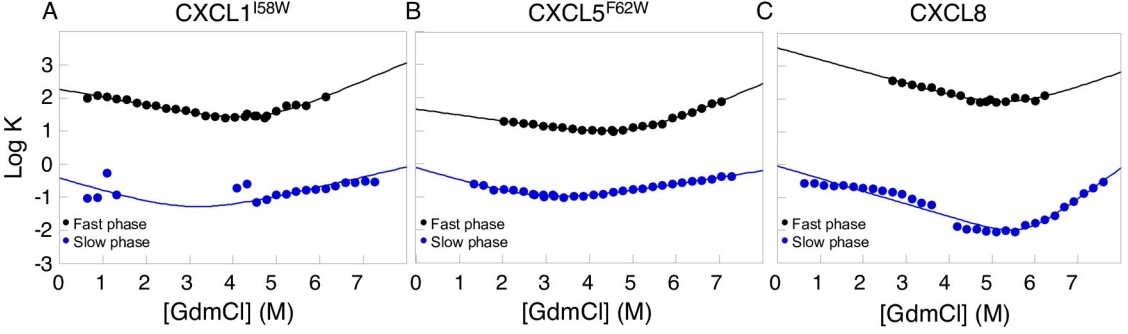

**Fig 3. Kinetics of CXC-ligands.** The two phases, a slow phase and a fast phase, indicates a three-state folding free energy landscape for A) CXCL1[I58W], B) CXCL5[F62W], and C) CXCL8.

**Table 2. Kinetics data for CXC-ligands in in 10 mM phosphate buffer at pH 7.4 at 25 ˚C.**

| | $\log k_f^{H_2O}$ | $m_f$ $M^{-1}$ | $\log k_u^{H_2O}$ | $m_u$ $M^{-1}$ | $\beta^{\ddagger}$ | MP M | $m_{D-N}$ $M^{-1}$ | $\log K_{D-N}^{H_2O}$ | $\Delta G$ kcal/mol |
|---|---|---|---|---|---|---|---|---|---|
| CXCL1[I58W] | 2.27 | -0.24 | -1.38 | 0.56 | 0.30 | 4.58 | 0.80 | 3.65 | 4.96 ± 0.69 |
| CXCL5[F62W] | 1.65 | -0.18 | -1.99 | 0.55 | 0.24 | 4.99 | 0.73 | 3.29 | 4.47 ± 2.58 |
| CXCL8 | 3.56 | -0.36 | -0.80 | 0.45 | 0.44 | 5.37 | 0.81 | 4.36 | 5.93 ± 1.83 |

*Fitted to a two-state equation of the fast phase.

**Table 3. Kinetics data for CXC-ligands in in 10 mM phosphate buffer at pH 7.4 at 25 ˚C.**

| | $\log k_f^{H_2O}$ | $m_f$ $M^{-1}$ | $\log k_u^{H_2O}$ | $m_u$ $M^{-1}$ | $\beta^{\ddagger}$ | MP M | $m_{D-N}$ $M^{-1}$ | $\log K_{D-N}^{H_2O}$ | $\Delta G$ kcal/mol |
|---|---|---|---|---|---|---|---|---|---|
| [1]CXCL1[I58W] | n.d. | n.d. | n.d. | n.d. | n.d. | n.d. | n.d. | n.d. | n.d. |
| CXCL5[F62W] | -0.12 | -0.39 | -1.85 | 0.21 | 0.66 | 2.90 | 2.90 | 1.97 | 2.69 ± 0.44 |
| CXCL8 | -0.04 | -0.39 | -7.67 | 0.95 | 0.29 | 5.37 | 5.75 | 7.71 | 10.49 ± 1.10 |

*Fitted to a two-state equation of the slow phase

[1]There is no steady slow phase at 2 µM, thus values not determined (n.d.).

the kinetics experiments. The results agree with the AUC data (S2 Fig in S1 File) and with previously published data [25–29]. The global stability for the slow phase of CXCL5[F62W] is in agreement with the thermodynamics data, i.e., $\Delta G$ = 2.69 ± 0.44 and 1.58 ± 3.55 kcal/mol, while CXCL8 has a higher kinetic stability of 10.49 ± 1.10 compared to 1.82 ± 5.27 kcal/mol. The observed contradiction may suggest heterogeneity in the sample with populations of more than one conformation rather than dimerization.

There is no observed slow phase for the refolding reaction for CXCL1[I58W], as dimeric species may not be formed at 2 µM. There is a signal observed for refolding at 30 µM for low denaturant concentration. There is no signal observed around the MP where the high GdmCl concentration inhibits dimer formation for CXCL1[I58W]. The asymmetric mixing may contribute to the dimer dissociation observed for the unfolding limb for CXCL1[I58W]. Denaturant was added to the refolding experiment to solve the complex kinetics observed. The effect of denaturant was only observed for CXCL8 while there was no effect on the refolding for CXCL1[I58W] and CXCL5[F62W] (S4 Fig in S1 File).

## Contact order

The contact order (CO) of a protein measures how sequentially close the amino acid residues are to each other within the protein's primary sequence, which provides information about the spatial organization of the three-dimensional structure [30]. Specifically, contact order is the average sequence separation between residues that are in contact in the folded protein. Plaxco *et al* demonstrated that the logarithm of the folding rate in water correlates with contact order [30, 31]. It is calculated as the average sequence distance between residues that form contacts in the native state divided by the total length of the polypeptide chain. CO quantifies the average number of amino acids that need to be traversed along the protein chain to connect two residues that are spatially close in the folded structure. Proteins with low contact order tend to have a more compact and globular structure, where neighboring residues in the primary sequence are also close in space, leading to faster refolding rates. The relative contact order

(RCO) is in agreement with the fast-folding rate constants of the fast phase, where CXCL8 folds faster than CXCL1$^{I58W}$ and CXCL5$^{F62W}$. The folding rate in water of CXCL8, CXCL1$^{I58W}$, and CXCL5$^{F62W}$ is $logk_f^{H_2O}$ is 3.56, 2.27, and 1.65 and the RCO is 0.112, 0.111, and 0.103 respectively [30, 31]. Although the $logk_f^{H_2O}$ and RCO follows the same trends, the change in refolding rate constant is larger than anticipated from the calculated RCO. The contribution from the hydrophobic effect, where hydrophobic amino acids are hidden in the core or the protein, may stabilize the transition state ensemble and thus speed up folding for CXCL8.

### The hydrophobic effect

The hydrophobic effect is the main driving force for folding where the system loses entropy as the nascent chain folds into its three-dimensional structure [32]. The final compact native state is stabilized by the gain in enthalpy on removal of non-polar groups from the contact with water [19, 33]. The hydrophobic effect is not particularly strong on its own, but the combination of all hydrophobic contacts within a protein is very strong. As a first approximation, hydrogen bonds and salt bridges are assumed to be nearly as strong in water as within a folded protein, and so, there is no major gain in terms of stability when forming these contacts. Thus, the hydrophobicity of proteins contributes to the overall global stability (ΔG) and refolding rate constants [34]. We utilized the cornett scale to calculate the hydrophobicity of the CXC-ligands, where hydrophobic amino acids have positive and hydrophilic negative numbers. Our result shows that CXCL5 is the most hydrophobic, i.e., 40.2, 69, and 53.4 for CXCL1, CXCL5, and CXCL8 respectively. This result contradicts the observed rate constant where CXCL5 folds slower than CXCL1 and CXCL8. Many bioinformatic tools can calculate hydrophobicity with varied results. The deviation is attributed to the fact that hydrophobicity is a relative quantity that depends on the environment and reference molecules used in the measurement. Taken together, we do not attribute the change in refolding rate constant to the hydrophobic effect.

### Activity assays

The CXC-ligands were purified from inclusion body purification utilizing a thioredoxin tag that was cleaved off rendering folded proteins with a molecular weight of 7,865 kDa for CXCL1, 8,357 kDa for CXCL5, and 8,385 kDa for CXCL8 (S1 Fig in S1 File). To ensure active ligand proteins from our inclusion body purification methodology, we conducted activity assays in HEK293-CXCR2 cells. CXCR2-mediated ERK phosphorylation was assessed in the presence of CXC-ligands at various concentrations, from 0 to 100 nM. All CXC-ligands induce a dose-dependent ERK phosphorylation upon activating CXCR2 (Fig 4). CXCL8 showed maximal ERK phosphorylation at a lower concentration (1 nM) relative to CXCL1 and CXCL5, which is in agreement with previously published data showing CXCL8 to be a stronger agonist for CXCR2 [25–29].

### Discussion

The promiscuous chemokine network with many possible oligomeric agonists/antagonists for the chemokine receptors, i.e., monomer, homodimer, and heterodimers, remains elusive. The key mechanism in cell migration is controlled by chemotaxis, which is the directed movement of cells in response to a gradient of chemokine concentration.

In this work, we utilize three of the CXCR2 ligands, namely CXCL1, CXCL5, and CXCL8. Activation of CXCR2 occurs through various oligomeric states of the CXC-ligands with different binding affinities. CXCL1 has a binding constant between 1–10 nM range indicating that

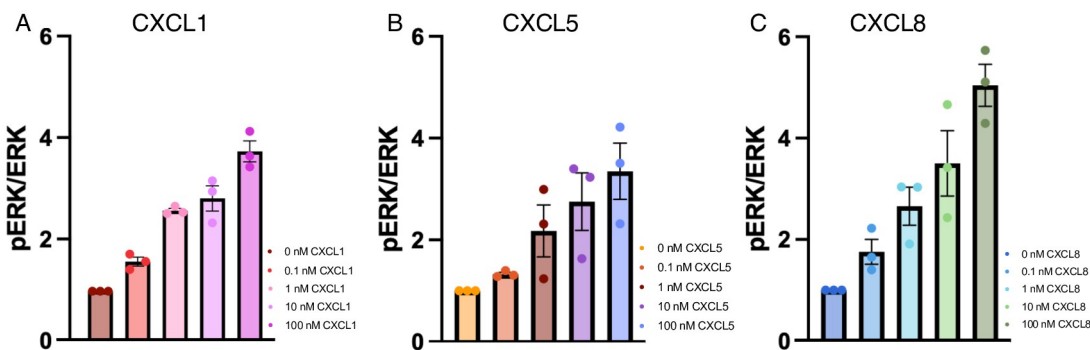

**Fig 4. Activity data for CXC-ligands.** CXCR2-mediated ERK phosphorylation was assessed in the presence of CXCL1, CXCL5, and CXCL8. All ligands induced a dose-dependent ERK phosphorylation (pERK) upon activating CXCR2 based on ANOVA analysis. CXCL8 showed maximal pERK at a lower concentration (1 nM) relative to CXCL1 and CXCL5.

CXCL1 is dominantly active as a monomer [25]. CXCL5 is a weaker agonist [26] and has a binding constant between 1–10 nM [27, 28]. CXCL8 is the most well-studied ligand for CXCR2 and can activate signaling both as a monomer and as a dimer with a binding constant between 10–100 nM [29]. It should be noted that the observed binding affinities vary as a function of pH, ionic strength, and buffer [21–23]. Taken together, CXCL8 has the strongest affinity for CXCR2 and CXCL5 the weakest interaction. This agrees with our activity data showing a dose dependance response of ERK phosphorylation, where CXCL8 shows maximal pERK at a lower concentration (1 nM), relative to CXCL1 and CXCL5.

## Structure and stability of CXC-ligands

The CD data of the CXC-ligands reveals small perturbations of the secondary structures in solution, which were larger than anticipated from the overlay of the crystal structures. The shifts are observed around 222 nm and 208 nm where CXCL8 displays a more helical spectrum than CXCL1 and CXCL5 (S1 Fig in S1 File). NMR studies of the homodimers and designed monomers show chemical shift perturbations in the α-helix, where the monomeric proteins show a truncated helix [35–37]. The small shifts seen in our CD spectrum may be affected by the population of monomers versus dimers in solution. To test the formation of homodimers at different concentrations, we conducted AUC experiments. The data shows that all three proteins are mainly dimeric at all concentrations tested (S2 Fig in S1 File). Various concentrations (2 to 100 μM) were tested for our equilibrium titrations (S3 Fig in S1 File). The lowest concentration tested is measured near the detection limit of the instrument, scattering the data, otherwise not observed at higher concentrations. Nonetheless, the concentration dependence observed in the equilibrium data exhibits three-state behavior with a monomeric intermediate state [38], as seen by the plateau introducing a bimodal concentration dependence at low denaturant concentrations for 2 μM protein. We hypothesize that CXC-ligands form a monomeric native state before dimerizing. We attribute the early TSE to the association/disassociation of the homodimer and the late phase at high denaturant concentrations to the folding of the monomeric proteins. This is further supported by our kinetics data and previously published data for other CC-ligands (CCL), such as CCL11 and CCL24 [24]. The kinetic data displays an atypical behavior with two separate phases, plotted as a slow and a fast phase (Fig 3). This is indicative of a three-state behavior, where an intermediate state accumulates rapidly during folding [39]. To test the concentration dependence for kinetics experiment, a final concentration between 2 to 30 μM was utilized (S4 Fig in S1 File). The data for high and low

protein concentrations coincide for CXCL1$^{I58W}$ and CXCL5 $^{F62W}$ for the unfolding limb of the slow phase but display a concentration dependance in the refolding limb. There is a downward shift in the unfolding limb observed for the slow phase of CXCL8 at high protein concentration, giving rise to a mismatch in the unfolding and refolding rates. To resolve the complex kinetics with mixed phases, we added 3.5 M GdmCl to the unfolding experiment, weakening the dimer association and shifting the unfolding limb of the slow phase to match the refolding limb (S4 Fig in S1 File). Taken together, an intermediate state populated on the folding free energy landscape would not display concentration dependence, while the formation of a homodimer is concentration dependent [38–40]. Therefore, we attribute the three-state behavior to the formation of a fully folded native$_{monomer}$.

## Dimer of CXC-ligands

The binding interfaces for the CXC-ligands are formed between residues in β-strand 1 of each monomer. Upon visual inspection in PyMOL, the contacts are mainly hydrophobic interactions, where the binding interface of the CXCL1 dimer forms between residues Q24-S30, V26-V28, V28-V26, and the CXCL5 dimer forms between residues S28-A34, L30-V32, V32-L30, A34-S28, and the CXCL8 dimer forms between residues K23-E29, L25-V27, V27-L25, E29-K23, from visual inspection using PyMOL. The dimer of CXCL1 forms through two hydrophobic interactions and one polar interaction between Q24-S30. The CXCL5 dimer expands the binding interface to include four contacts, two hydrophobic interactions and two hydrogen bonds between S28-A34. The CXCL8 dimer forms a binding interface with two hydrophobic interactions in-between two ionic contacts between K23-E29 of each monomer. A larger binding interface is seen for CXCL8 affecting the global stability as shown with the two-state fit of the equilibrium titrations, ΔG = 7.07 ± 0.10 kcal/mol and the slow phase in the kinetics data ΔG = 10.49 ± 1.10 kcal/mol (S1 Table in S1 File and Table 3). This explains why CXCL1 forms the weakest dimer, with an equilibrium constant of ~20 μM and exists as a monomer *in vivo* [25], CXCL5 exists both as monomers and dimers with a monomer-dimer equilibrium constant of ~1 μM [27, 28], while CXCL8 exists both as a monomer and dimer with a monomer-dimer equilibrium constant of ~0.1 μM [23]. Again, it should be noted that the observed binding affinities vary as a function of pH, ionic strength, and buffer [21–23].

## Conclusions

Insight on the molecular details of the chemokine signaling pathways may aid in the development of therapeutics to target the CXCR2. CXCR2 is expressed on the surface of various immune cells, including neutrophils, monocytes, and certain subsets of T cells. Biophysical characterization is the first step to understand the formation of the oligomeric states that can activate chemokine receptors. Our data show that CXC-ligands fold through a three-state mechanism, where a folded monomeric state is on pathway to the active dimer. This is also observed for the CC-chemokines [24], suggesting that the folding mechanism is conserved across the chemokine family. Furthermore, we hypothesize that sequence diversity through evolution is introduced for specificity to control biological activity. In this study, we show that CXCL8 forms the strongest homodimer with the highest biological activity, in agreement with previously published results of CXCL8 being the strongest agonist for CXCR2. Further research on CXC monomeric proteins is required to unravel the complex kinetics and the molecular details controlling this promiscuous chemokine network.

## Supporting information

**S1 File.**
(ZIP)

## Acknowledgments

We want to thank Dr. Krishna Rajarathnam from the University of Texas Medical Branch at Galveston and the Department of Biochemistry and Molecular Biology for providing the plasmid for CXCL5. We also want to thank National Magnetic Resonance Facility at Madison (NMRFAM) for collecting the NMR spectrum in the S1 File.

## Author Contributions

**Conceptualization:** Ellinor Haglund.

**Data curation:** Patrick Martin, Emily A. Kurth, David Budean, Nathalie Momplaisir, Elaine Qu, Chad A. Brautigam.

**Formal analysis:** Patrick Martin, Emily A. Kurth, David Budean, Nathalie Momplaisir, Elaine Qu, Jennifer M. Simien, Grace E. Orellana, Chad A. Brautigam, Alan V. Smrcka, Ellinor Haglund.

**Funding acquisition:** Alan V. Smrcka, Ellinor Haglund.

**Investigation:** Alan V. Smrcka, Ellinor Haglund.

**Methodology:** Patrick Martin, Emily A. Kurth, David Budean, Nathalie Momplaisir, Elaine Qu, Jennifer M. Simien, Grace E. Orellana, Chad A. Brautigam, Ellinor Haglund.

**Supervision:** Ellinor Haglund.

**Writing – original draft:** Ellinor Haglund.

**Writing – review & editing:** Patrick Martin, Emily A. Kurth, David Budean, Nathalie Momplaisir, Jennifer M. Simien, Grace E. Orellana, Chad A. Brautigam, Alan V. Smrcka.

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
