## [Decision Letter · Decision Letter 0]

5 Dec 2023

PONE-D-23-32425Biophysical Characterization of the CXC Chemokine Receptor 2 LigandsPLOS ONE

Dear Dr. Haglund,

Thank you for submitting your manuscript to PLOS ONE. After careful consideration, we feel that it has merit but does not fully meet PLOS ONE’s publication criteria as it currently stands. Therefore, we invite you to submit a revised version of the manuscript that addresses the major and minor raised during the review process by both reviewers.

We look forward to receiving your revised manuscript.

Kind regards,

Samithamby Jeyaseelan (Jey), DVM, PHD

Academic Editor

PLOS ONE

Journal Requirements:

National Science Foundation (NSF) award number CHE2145906

the Hawaii Community Foundation (HCF) award number HCF40846 (013357-00002)

We want to thank Dr. Krishna Rajarathnam from the University of Texas Medical Branch at Galveston and the department of biochemistry and molecular biology for providing the plasmid for CXCL5. The research reported by the Haglund group is supported by the National Science Foundation award number CHE2145906 and the Hawaii Community Foundation award number HCF40846 (013357-00002). The research reported by the Smrcka group is supported by the National Institute of Health grant number R35GM127303.

National Science Foundation (NSF) award number CHE2145906

the Hawaii Community Foundation (HCF) award number HCF40846 (013357-00002)

Reviewers' comments:

Reviewer's Responses to Questions

**Comments to the Author**

1. Is the manuscript technically sound, and do the data support the conclusions?

Reviewer #1: Yes

Reviewer #2: Yes

2. Has the statistical analysis been performed appropriately and rigorously? 

Reviewer #1: No

Reviewer #2: N/A

3. Have the authors made all data underlying the findings in their manuscript fully available?

Reviewer #1: Yes

Reviewer #2: Yes

4. Is the manuscript presented in an intelligible fashion and written in standard English?

Reviewer #1: Yes

Reviewer #2: Yes

5. Review Comments to the Author

Reviewer #1: In this manuscript, Martin and his colleagues conducted an investigation into the homodimers of CXCR2 cognate ligands, namely CXCL1, CXCL5, and CXCL8. The findings revealed that these three ligands exhibit a significant degree of structural similarity, despite having a sequence identity of less than 25%. This sequence diversity has led to the development of varying binding affinities and stabilities for the CXC-ligands, consequently resulting in a diverse range of agonist and antagonist behaviors. The folding process of CXC-ligands follows a three-state mechanism, wherein they first adopt a folded monomeric state before transitioning into an active dimeric configuration. Overall, the topic is interesting. Nevertheless, there are certain concerns that need to be addressed in the current version of the manuscript.

1. In the Introduction section, the purpose of the work is not clearly defined, and crucial information concerning the chemokine receptor and ligand is absent. Additional details pertaining to their structural aspects, functions, and binding sites should be elaborated upon. Furthermore, citations in the Introduction section should be incorporated to support the presented information.

2. In the first two paragraphs of the Results section, it is somewhat unclear what specific results were derived from this study. These paragraphs appear to be more aligned with providing background knowledge and might be better suited for inclusion in the Introduction section.

3. Did both the contact order and stability of CXCL1, CXCL5, and CXCL8 play a critical role in determining their folding rates when interacting with CXC chemokine receptors and ligands?

4. In Figure S1, the absorbance values (y-axis) are missing.

5. Figure 4 would benefit from the inclusion of representative bands for pERK/ERK to provide stronger support for the conclusions regarding changes in activity.

6. Statistical analysis for quantitative data is notably absent.

7. On page 1, the correct term for the CXC chemokine receptor 2 (CXCL2) should be "CXCR2."

Reviewer #2: Patrick Martin et al. conducted a study on the "Biophysical Characterization of the CXC Chemokine Receptor 2 Ligands" to investigate the homodimers of three CXCR2 cognate ligands: CXCL1, CXCL5, and CXCL8. Despite having high structural integrity, these ligands have low sequence identity. The study found that CXCL8 forms the strongest homodimer with the most increased biological activity, making it the most potent agonist for CXCR2. Additionally, activity assays revealed that CXCL8 showed maximal ERK phosphorylation at a lower concentration than CXCL1 and CXCL5. Therefore, the researchers concluded that CXCL8 is the most effective ligand for CXCR2.

The experiments were planned and presented effectively, highlighting the various strengths of the manuscript.

I have minor suggestions:

a. What are A293 cells? Include the full name and source.

b. Page 3 - “The ligands were strategically selected from sequence evolution representing different branches of the phylogenetic tree (Figure 1A).” Elaborate on what strategic selection means.

c. Including the chemokine ligand names in the titles of figures would increase clarity (for example, for figure 2A you can title the plot “CXCL1” or something similar.

d. Perhaps a more detailed discussion of the process identifying the three-state mechanism and its implications could add to readers’ understanding.

e. Please conduct a careful grammar check throughout for minor errors.

f. Please introduce all acronyms when they first appear if not already done.

6. PLOS authors have the option to publish the peer review history of their article (what does this mean?). If published, this will include your full peer review and any attached files.

Reviewer #1: No

Reviewer #2: No

---

## [Author Response · Author response to Decision Letter 0]

22 Jan 2024

Response to Reviewers, PONE-D-23-32425

Biophysical Characterization of the CXC Chemokine Receptor 2 Ligands

PLOS ONE

We thank the editor and the reviewers for their careful review of our manuscript entitled “Biophysical Characterization of the CXC Chemokine Receptor 2 Ligands” by Martin et al. We are glad to hear the positive response regarding our research topic and the interest in chemokine ligands. We agree with the reviewer’s feedback and, thus, have made suggested changes to the manuscript. We believe that the reviewer feedback has strengthen our manuscript. 

We have prepared a point-by-point response to the reviewers' concerns along with a detailed description of the changes that were made in response to these concerns. Attach are the highlighted manuscript including the changes in the manuscript and the supporting information.

Thank you for your time and consideration!

With best regards,

Ellinor 

REVIEWER 1

In this manuscript, Martin and his colleagues conducted an investigation into the homodimers of CXCR2 cognate ligands, namely CXCL1, CXCL5, and CXCL8. The findings revealed that these three ligands exhibit a significant degree of structural similarity, despite having a sequence identity of less than 25%. This sequence diversity has led to the development of varying binding affinities and stabilities for the CXC-ligands, consequently resulting in a diverse range of agonist and antagonist behaviors. The folding process of CXC-ligands follows a three-state mechanism, wherein they first adopt a folded monomeric state before transitioning into an active dimeric configuration. Overall, the topic is interesting. Nevertheless, there are certain concerns that need to be addressed in the current version of the manuscript.

- Author reply: We thank reviewer one for taking their time to reviewer our manuscript and we are pleased to hear that Reviewer 1 likes our paper.

1. In the Introduction section, the purpose of the work is not clearly defined, and crucial information concerning the chemokine receptor and ligand is absent. Additional details pertaining to their structural aspects, functions, and binding sites should be elaborated upon. Furthermore, citations in the Introduction section should be incorporated to support the presented information.

Author reply: We thank the reviewer for addressing the lack of purpose described in the introduction. We have moved the first two paragraphs from the RESULTS section as suggested in comment 2 below. We have also added citations to the introduction.

 Zlotnik A, Yoshie O. Chemokines: a new classification system and their role in immunity. Immunity. 2000;12(2):121-7.

 Buck MD, Sowell RT, Kaech SM, Pearce EL. Metabolic Instruction of Immunity. Cell. 2017;169(4):570-86.

 Ganeshan K, Chawla A. Metabolic regulation of immune responses. Annu Rev Immunol. 2014;32:609-34.

 O'Neill LA, Kishton RJ, Rathmell J. A guide to immunometabolism for immunologists. Nat Rev Immunol. 2016;16(9):553-65.

 Hughes CE, Nibbs RJB. A guide to chemokines and their receptors. FEBS J. 2018;285(16):2944-71.

 Korbecki J, Kupnicka P, Chlubek M, Goracy J, Gutowska I, Baranowska-Bosiacka I. CXCR2 Receptor: Regulation of Expression, Signal Transduction, and Involvement in Cancer. Int J Mol Sci. 2022;23(4).

 Rajagopalan L, Rajarathnam K. Structural basis of chemokine receptor function--a model for binding affinity and ligand selectivity. Biosci Rep. 2006;26(5):325-39.

2. In the first two paragraphs of the Results section, it is somewhat unclear what specific results were derived from this study. These paragraphs appear to be more aligned with providing background knowledge and might be better suited for inclusion in the Introduction section.

Author reply: We thank the reviewer for the suggestion to move the first two paragraphs from the RESULTS section to the introduction, and added the following section to the RESULTS section on page 7: Understanding the three-dimensional structure and the thermodynamic and kinetic behavior of proteins are essential in understanding protein-protein interaction. The first structure of CXC-chemokines was solved about 30 years ago. However, the biophysical character is not known for CXC-ligands. In this work, we investigate the molecular details controlling protein-protein interaction to understand the structural integrity controlling the oligomeric state of CXC-chemokines.

3. Did both the contact order and stability of CXCL1, CXCL5, and CXCL8 play a critical role in determining their folding rates when interacting with CXC chemokine receptors and ligands?

Author reply: The rate constant for folding was observed in the absence of the receptor. The relative contact order (RCO) for proteins correlates with the observed rate constants for folding. In the case of the CXC-chemokines, the folding rate and relative contact order are in agreement between the chemokines, i.e., CXCL8 folds faster than CXCLI58W and CXCL5F62W. The RCO follows the same trend where CXCL8 has a higher RCO than CXCLI58W and CXCL5F62W. However, upon closer examination, the RCO for the CXC-ligands folds slower than anticipated from the RCO ln(K) plot by Plaxco et al (1998). Furthermore, the hydrophobic effect dose not significantly affect the refolding rate constants. We have added a section to discuss this on page 10: The relative contact order (RCO) is in agreement with the fast-folding rate constants of the fast phase, where CXCL8 folds faster than CXCL1I58W and CXCL5F62W, i.e., 〖logk〗_f^(H_2 O)is 3.56, 2.27, and 1.65 and the RCO is 0.112, 0.111, and 0.103 respectively (30, 31). Although the 〖logk〗_f^(H_2 O) and RCO follows the same trends, the change in refolding rate constant is larger than anticipated from the calculated RCO. The contribution from the hydrophobic effect, where hydrophobic amino acids are hidden in the core or the protein, may stabilize the transition state ensemble and thus speed up folding for CXCL8. 

We added a section discussing the hydrophobic effect on page 10-11:

The hydrophobic effect. The hydrophobic effect is the main driving force for folding where the system loses entropy as the nascent chain folds into its three-dimensional structure (32). The final compact native state is stabilized by the gain in enthalpy on removal of non-polar groups from the contact with water (19, 33). The hydrophobic effect is not particularly strong on its own, but the combination of all hydrophobic contacts within a protein is very strong. As a first approximation, hydrogen bonds and salt bridges are assumed to be nearly as strong in water as within a folded protein, and so, there is no major gain in terms of stability when forming these contacts. Thus, the hydrophobicity of proteins contributes to the overall global stability (�G) and refolding rate constants (34). We utilized the cornett scale to calculate the hydrophobicity of the CXC-ligands, where hydrophobic amino acids have positive and hydrophilic negative numbers. Our result shows that CXCL5 is the most hydrophobic, i.e., 40.2, 69, and 53.4 for CXCL1, CXCL5, and CXCL8 respectively. This result contradicts the observed rate constant where CXCL5 folds slower than CXCL1 and CXCL8. Many bioinformatic tools can calculate the hydrophobicity with varied results. The deviation is attributed to the fact that hydrophobicity is a relative quantity that depends on the environment and reference molecules used in the measurement. Taken together, we do not attribute the change in refolding rate constant to the hydrophobic effect. 

4. In Figure S1, the absorbance values (y-axis) are missing.

Author reply: We thank the reviewer for noticing that the values for the y-axis are missing. We have updated Figure S1, on page 2. 

5. Figure 4 would benefit from the inclusion of representative bands for pERK/ERK to provide stronger support for the conclusions regarding changes in activity. Statistical analysis for quantitative data is notably absent.

Author reply: We thank the reviewer for the comment regarding the pERK/ERK expression and the analysis methodology for our activity data. We utilized the ANOVA statistical analysis method. We have updated the figure legend for Figure 4 and Figure S5 to include: “Figure 4. Activity data for CXC-ligands. CXCR2-mediated ERK phosphorylation was assessed in the presence of CXCL1, CXCL5, and CXCL8. All ligands induced a dose-dependent ERK phosphorylation (pERK) upon activating CXCR2 based on ANOVA analysis. CXCL8 showed maximal pERK at a lower concentration (1 nM) relative to CXCL1 and CXCL5.”

“Figure S5: The tryptophan variants of CXCL1 and CXCL5. A) CXCL1 wild-type versus CXCL1I58W. B) CXCL5 wild-type versus CXCL5F62W. The amino acid substitution introduces a fluorescent probe. Substituting isoleucine in CXCL1 and phenylalanine in CXCL5 at the N-terminus of the �-helix stabilizes the helix and thus increases the �G (Table S2). All proteins induce a comparable dose-dependent CXCR2-mediated ERK phosphorylation. The pseudo wild-type proteins cause a minor shift in the dose response, where CXCL1I58W decreases the response and CXCL5F62W increases the response, but maximal efficacy in mediating ERK phosphorylation relative to the wild-type proteins based on ANOVA analysis. 

7. On page 1, the correct term for the CXC chemokine receptor 2 (CXCL2) should be "CXCR2."- Author reply: We thank the reviewer for pointing out our typo. We have changed the name of the receptor to CXCR2 on page 1. 

REVIEWER 2

Patrick Martin et al. conducted a study on the "Biophysical Characterization of the CXC Chemokine Receptor 2 Ligands" to investigate the homodimers of three CXCR2 cognate ligands: CXCL1, CXCL5, and CXCL8. Despite having high structural integrity, these ligands have low sequence identity. The study found that CXCL8 forms the strongest homodimer with the most increased biological activity, making it the most potent agonist for CXCR2. Additionally, activity assays revealed that CXCL8 showed maximal ERK phosphorylation at a lower concentration than CXCL1 and CXCL5. Therefore, the researchers concluded that CXCL8 is the most effective ligand for CXCR2. The experiments were planned and presented effectively, highlighting the various strengths of the manuscript.

- Author reply: We thank reviewer one for taking their time to reviewer our manuscript and we are pleased to hear that Reviewer 1 likes our paper.

a. What are A293 cells? Include the full name and source.

- Author reply: We thank the reviewer for pointing out that we missed typing the full name and source for our human cell liens. The cells used are HEK293 cells. We have added the full name to the following sentence on page 5: HEK293 cells (CRL-1573, ATCC) were maintained in DMEM (10013 CV, Corning) supplemented with 10% FBS (10437028, Gibco) and 1% Pen/Strep (15140122, Gibco) at 37 °C with 5% CO2.

b. Page 3 - “The ligands were strategically selected from sequence evolution representing different branches of the phylogenetic tree (Figure 1A).” Elaborate on what strategic selection means.

- Author reply: We thank the reviewer for the feedback and have added the following section to clarify how we “strategically selected” our CXCR2 ligands: “The CXCR2 receptor has seven cognate ligands that may interact and initiate a cell signaling response. CXCL1, CXCL5, and CXCL8 were selected to represent the most diverse ligands for CXCR2 with a common ancestor.”

c. Including the chemokine ligand names in the titles of figures would increase clarity (for example, for figure 2A you can title the plot “CXCL1” or something similar.

- Author reply: We thank the reviewer for pointing out that a figure legend may be helpful. We have updated all figures to clarify what data is from CXCL1, CXCL5, or CXCL8. 

d. Perhaps a more detailed discussion of the process identifying the three-state mechanism and its implications could add to readers’ understanding.

- Author reply: We thank the reviewer for the concern regarding the three-state mechanism discussion. We have added a section to the Discussion section, “Structure and stability of CXC-ligands”, on page 12 to clarify: The data for high and low protein concentrations coincide for CXCL1I58W and CXCL5 F62W for the unfolding limb of the slow phase but display a concentration dependance in the refolding limb. There is a downward shift in the unfolding limb observed for the slow phase of CXCL8 at high protein concentration, giving rise to a mismatch in the unfolding and refolding rates. To resolve the complex kinetics, with mixed phases we added 3.5 M GdmCl to the unfolding experiment, weakening the dimer association and shifting the unfolding limb of the slow phase to match the refolding limb (Figure S4). Taken together, an intermediate state populated on the folding free energy landscape would not display concentration dependence, while the formation of a homodimer is concentration dependent (38-40) . Therefore, we attribute the three-state behavior to the formation of a fully folded nativemonomer.

e. Please conduct a careful grammar check throughout for minor errors.

- Author reply: We thank the reviewer for pointing out that the manuscript has grammar errors. We have thoroughly edited the manuscript fixing typos and grammar, see manuscript with track changes. 

f. Please introduce all acronyms when they first appear if not already done.

- Author reply: We thank the reviewer for pointing out we missed to introduce acronyms the first time we used it. We have therefore updated the manuscript.

---

## [Decision Letter · Decision Letter 1]

25 Jan 2024

Biophysical Characterization of the CXC Chemokine Receptor 2 Ligands

PONE-D-23-32425R1

Dear Dr. Haglund,

We’re pleased to inform you that your manuscript has been judged scientifically suitable for publication and will be formally accepted for publication once it meets all outstanding technical requirements.

Kind regards,

Samithamby Jeyaseelan, DVM, PHD

Academic Editor

PLOS ONE

Additional Editor Comments (optional):

The revisions have been successfully addressed.

Reviewers' comments:

Reviewer's Responses to Questions

**Comments to the Author**

1. If the authors have adequately addressed your comments raised in a previous round of review and you feel that this manuscript is now acceptable for publication, you may indicate that here to bypass the “Comments to the Author” section, enter your conflict of interest statement in the “Confidential to Editor” section, and submit your "Accept" recommendation.

Reviewer #1: All comments have been addressed

2. Is the manuscript technically sound, and do the data support the conclusions?

Reviewer #1: Yes

3. Has the statistical analysis been performed appropriately and rigorously? 

Reviewer #1: Yes

4. Have the authors made all data underlying the findings in their manuscript fully available?

Reviewer #1: Yes

5. Is the manuscript presented in an intelligible fashion and written in standard English?

Reviewer #1: Yes

6. Review Comments to the Author

Reviewer #1: The authors have adequately addressed the comments made by the reviewers in the revised version of the manuscript. I have no further comments.

7. PLOS authors have the option to publish the peer review history of their article (what does this mean?). If published, this will include your full peer review and any attached files.

Reviewer #1: No

---

## [Editor Report · Acceptance letter]

26 Feb 2024

PONE-D-23-32425R1 

PLOS ONE

Dear Dr. Haglund, 

I'm pleased to inform you that your manuscript has been deemed suitable for publication in PLOS ONE. Congratulations! Your manuscript is now being handed over to our production team.

Kind regards, 

on behalf of

Dr. Samithamby Jeyaseelan 

Academic Editor

PLOS ONE